# The Expression and Functionality of CB_1_R-NMDAR Complexes Are Decreased in A Parkinson’s Disease Model

**DOI:** 10.3390/ijms25053021

**Published:** 2024-03-05

**Authors:** Irene Reyes-Resina, Jaume Lillo, Iu Raïch, Joan Biel Rebassa, Gemma Navarro

**Affiliations:** 1Department of Biochemistry and Physiology, School of Pharmacy and Food Sciences, Universitat de Barcelona, 08028 Barcelona, Spain; iraichpa7@ub.edu (I.R.); jrebaspa7@alumnes.ub.edu (J.B.R.); 2Network Center for Biomedical Research in Neurodegenerative Diseases, CiberNed, Spanish National Health Institute Carlos III, Av. Monforte de Lemos, 3-5, 28029 Madrid, Spain; jaumelillo@ub.edu; 3Institut de Neurociències UB, Campus Mundet, Passeig de la Vall d’Hebron 171, 08035 Barcelona, Spain; 4Molecular Neurobiology Laboratory, Department Biochemistry and Molecular Biomedicine, Facultat de Biologia, Universitat de Barcelona, 08028 Barcelona, Spain

**Keywords:** Parkinson’s disease, cannabinoid receptor 1, NMDA receptor, receptor heteromer, α-synuclein

## Abstract

One of the hallmarks of Parkinson’s disease (PD) is the alteration in the expression and function of NMDA receptor (NMDAR) and cannabinoid receptor 1 (CB_1_R). The presence of CB_1_R-NMDAR complexes has been described in neuronal primary cultures. The activation of CB_1_R in CB_1_R-NMDAR complexes was suggested to counteract the detrimental NMDAR overactivation in an AD mice model. Thus, we aimed to explore the role of this receptor complex in PD. By using Bioluminescence Resonance Energy Transfer (BRET) assay, it was demonstrated that α-synuclein induces a reorganization of the CB_1_R-NMDAR complex in transfected HEK-293T cells. Moreover, α-synuclein treatment induced a decrease in the cAMP and MAP kinase (MAPK) signaling of both CB_1_R and NMDAR not only in transfected cells but also in neuronal primary cultures. Finally, the interaction between CB_1_R and NMDAR was studied by Proximity Ligation Assay (PLA) in neuronal primary cultures, where it was observed that the expression of CB_1_R-NMDAR complexes was decreased upon α-synuclein treatment. These results point to a role of CB_1_R-NMDAR complexes as a new therapeutic target in Parkinson’s disease.

## 1. Introduction

Parkinson’s disease is the second most common neurodegenerative disorder, affecting 2–3% of the population above 65 years old [1]. It courses with a degeneration of the indirect pathway of the basal ganglia in a chronical and progressive way, with a higher prevalence in males than in females [2]. PD is characterized by the loss of the dopaminergic neurons of the substantia nigra pars compacta, whose axons project to the striatum. This lack of dopamine in the striatum produces a dysregulation of the direct and indirect pathways, resulting in symptoms such as bradykinesia, rigidity, tremor, and postural instability [1]. The mechanism leading to this nigral dopaminergic neuronal degeneration remains unknown. Most PD cases are sporadic, being associated with variants of genes, which, through interaction with environmental toxins, can influence the susceptibility to PD [3]. The familial form of the disease accounts for only 14% of PD cases, and it has been linked to mutations in different genes, such as α-synuclein, parkin, DJ-1, PINK1, and LRRK2 [4].

One of the hallmarks of a PD brain are Lewy bodies, which are intracellular inclusions composed of aggregates of misfolded α-synuclein (α-syn) present in the remaining dopaminergic neurons [5]. α-syn is a presynaptic protein whose function is not well understood, although it has been described to participate in the biosynthesis and liberation of neurotransmitter vesicles by promoting SNARE-complex assembly during vesicle docking and fusion steps [6]. The deposition of α-syn aggregates occurs in the brain of patients with both familial and sporadic PD. In the familial form of the disease, the misfolding of α-syn is due to mutations in the SNCA gene, while in the sporadic form of PD, it is due to other risk factors [7,8]. Once the protein is misfolded, it aggregates, forming oligomers, which will further aggregate in the form of fibrils, which will finally create Lewy bodies [9]. These pathogenic aggregates of α-syn are transmitted between cells, contributing to PD propagation throughout the brain [10]. Further evidence of the implication of α-syn in the pathology comes from the fact that mutations and polymorphisms in the α-syn gene are related to the different forms of Parkinson’s disease [4,7], and both in vitro and in vivo models of α-syn overexpression have shown α-syn aggregation and neuronal loss [11,12,13].

Another feature of PD is the alteration in the expression and the activation of N-methyl-D-aspartate receptors (NMDARs) [14]. NMDARs are ionotropic glutamate receptors composed of four subunits, and in humans they are formed usually by two obligatory NR1 subunits plus two NR2A or NR2B subunits [15]. Upon activation, these receptors allow Ca^2+^ entry into the cell [15]. NMDARs are abundant in the postsynapse, as they have an important role in excitatory synaptic transmission by regulating mechanisms such as long-term potentiation (LTP) and long-term-depression (LTD) [16,17]. Overexpression of some NMDAR subunits has been found in animal models of PD [18], and the ratios between the different NMDAR subunits are altered in PD patients and in animal models of the disease [14,19]. Also, the NMDAR phosphorylation state is altered in rodent PD models consisting of unilateral nigrostriatal dopamine system ablation with 6-hydroxydopamine [14,20]. Furthermore, NMDAR antagonists such as MK-801, or negative allosteric modulators such as radiprodil, have been shown to improve motor symptoms in PD [21,22].

NMDARs are known to be able to interact with G protein-coupled receptors (GPCRs) that can be involved in the regulation of NMDAR function. For instance, trimeric complexes formed between NMDAR, dopamine D_1_ receptors, and histamine H_3_ receptors [23], or dimeric complexes formed between NMDAR and adenosine A_2A_ receptors [24], have been described to be potential targets for neurodegenerative diseases. We have previously described that NMDAR can also interact with cannabinoid receptors CB_1_ [25] and CB_2_ [26]. CB_1_R is the most abundant GPCR in the Central Nervous System (CNS), and it is mainly localized at the presynapse, where it regulates neurotransmission by inhibiting the release of neurotransmitter vesicles [27]. In CB_1_R-NMDAR complexes, CB_1_R regulates the function of NMDARs, as CB_1_R agonists have been described to induce a blockade of NMDAR signaling [25]. This is very interesting from a therapeutic point of view because cannabinoid agonists would not only provide the beneficial neuroprotective effects, but they also could counteract NMDAR overactivation, reducing NMDAR-induced excitotoxicity.

Nowadays, there is not a cure for Parkinson’s disease. The most common therapy consists of the dopamine precursor levodopa; however, it produces secondary effects, such as the abnormal involuntary movements known as dyskinesias [28]. Regarding the research on new therapeutic strategies for the treatment of this pathology, cannabinoid receptors have become an interesting target in PD. On the one hand, it has been shown that, in PD patients and in animal models of PD, there is a decrease in CB_1_R expression levels [29,30], as well as an increase in the levels of the endocannabinoids anandamide (AEA) [31,32] and 2-arachidonoyl glycerol (2-AG) [33]. On the other hand, another interesting feature of cannabinoids in the field of neurodegenerative diseases is their well-described neuroprotective potential [34,35]. Post-mortem analysis of PD brains shows the presence of activated microglia surrounding the dopaminergic neurons [36], indicating that accumulation of synuclein fibrils causes microglial activation. Indeed, it is known that neurodegenerative diseases such as Parkinson’s and Alzheimer’s diseases course with neuroinflammation [37,38].

Both Parkinson’s and Alzheimer’s diseases are neurodegenerative diseases where cannabinoids have shown a neuroprotective potential [35]. The fact that, in neurons of an animal model of Alzheimer’s disease, the expression of the CB_1_R-NMDAR complex was increased suggests that this receptor heteromer could have a role as a therapeutic target in neurodegenerative diseases [25]. This led us to think that the CB_1_R-NMDAR complex could also be important in Parkinson’s disease. Thus, in view of the necessity of new treatments for this pathology, our aim was to study the formation and function of CB_1_R-NMDAR complexes in an in vitro PD model consisting of neuronal primary cultures treated with α-syn, trying to show if CB_1_R-NMDAR complexes could have a role as therapeutic target in Parkinson’s disease.

## 2. Results

### 2.1. α-Syn Induces a Reorganization in the CB_1_R-NMDAR Complex Structure

To study the role of the CB_1_R-NMDAR heteromer in a parkinsonian context, we first determined whether α-syn had any effect on the receptors’ expression localization by performing an immunocytochemical assay in a heterologous expression system. HEK-293T cells expressing either CB_1_R fused to the fluorescent protein YFP (CB_1_R-YFP) or the NR1 subunit of NMDAR fused to the bioluminescent protein Renilla luciferase (NR1-Rluc) plus the NR2 subunit of NMDAR were treated with 10 μg/L of human alpha-synuclein (α-syn) fibrils or vehicle for 48 h. NMDAR was detected by an anti-Rluc primary antibody plus a secondary antibody labeled with Cy3. CB_1_R was detected by reading YFP’s own fluorescence. In the absence of α-syn fibrils, cells expressing only CB_1_R or NMDAR showed receptor localizations at the cell membrane level (Figure 1A). When cells co-expressed both receptors, the co-localization of CB_1_R and NMDAR was detected at the plasma membrane (Figure 1A), as previously described [25]. The presence of α-syn was detected by immunocytochemistry with an anti-human alpha synuclein antibody. Compared to the cells treated with the vehicle that show no signal (Figure 1B), the treated cells showed small red dots, demonstrating that HEK-293T cells are able to incorporate α-syn fibrils. When HEK-293T cells were pre-treated with the α-syn fibrils, neither the localization of CB_1_R and NMDAR nor their co-localization at the plasma membrane was affected (Figure 1A).

To assess if CB_1_ and NMDA receptors can directly interact, a Bioluminescence Resonance Energy Transfer (BRET) assay was performed. This technique allows us to determine protein–protein interactions at distances of maximum 10 nm. In HEK-293T cells expressing constant amounts of NR1-Rluc plus NR2B subunits of NMDAR and increasing amounts of CB_1_R-YFP, a saturation curve was obtained, indicating a specific interaction between CB_1_R and NMDAR (Figure 1C), with a BRET_max_ of 52.47 ± 14 mBU (milli BRET units) and BRET_50_ of 41.87 ± 15. When the interaction between CB_1_R and NMDAR was tested in the presence of α-syn fibrils, the BRET signal increased (Figure 1C), with a BRET_max_ of 90.2 ± 17 mBU (milli BRET units) and BRET_50_ of 57.66 ± 20. These results indicate that α-syn fibrils not only do not disrupt CB_1_R-NMDAR interaction but also may cause a higher number of complexes or a protein reorganization at the membrane that changes the orientation of the Rluc and YFP, allowing for a better transfer of energy between them.

However, in HEK-293T cells expressing constant amounts of NR1-Rluc plus NR2B subunits of NMDAR, together with increasing amounts of the ghrelin receptor GHSR_1A_ fused to YFP, a linear signal was obtained (Figure 1C), indicating a lack of interaction between NMDAR and GHSR_1A_.

### 2.2. CB_1_R Signaling Is Decreased by Treatment with α-Syn Fibrils in a Heterologous System

Next, we wanted to determine if α-syn fibrils were able to affect the functionality of the CB_1_R-NMDAR complex. First, we assessed if the fibrils had any effect on the signaling of either CB_1_R or NMDAR. As CB_1_R is mainly coupled to Gi protein, causing a decrease in intracellular cAMP levels upon receptor activation, the intracellular cAMP concentration was measured. In HEK-293T cells expressing CB_1_R, the treatment with the selective CB_1_R agonist ACEA was able to decrease the rise in cAMP levels induced by pre-treatment with forskolin (Figure 2A). When cells were pre-incubated with CB_1_R antagonist rimonabant (SR1416A), it was able to block the effect of ACEA. In the case of HEK-293T cells expressing NMDAR (NR1 plus NR2B subunits), the treatment with the agonist NMDA did not modify cAMP levels (Figure 2C), as previously described [26]. This result was expected due to the fact that NMDAR is an ion channel, and thus it is not coupled to G proteins, so its activation is not leading to cAMP-level modification [15]. When cells expressing CB_1_R were pre-treated with α-syn fibrils, ACEA treatment was not able to induce a decrease in cAMP levels (Figure 2B). In cells expressing NMDAR, the results observed upon α-syn fibrils were similar to those observed in cells treated with the vehicle (Figure 2D). Thus, α-syn fibrils cause a reduction in CB_1_R activation in HEK-293T cells.

As NMDAR activation causes Ca^2+^ entry to the cell, we then tested if α-syn fibrils affected intracellular calcium accumulation upon receptor activation. In HEK-293T cells expressing CB_1_R, treatment with the CB_1_R selective agonist ACEA was not able to move intracellular calcium (Figure 2E), which was not surprising, as CB_1_R is mainly coupled to Gi and not to Gq protein [39]. However, when HEK-293T cells expressing NMDAR were treated with the agonist NMDA, a rise in intracellular Ca^2+^ was detected, which was counteracted upon pre-treatment with NMDAR antagonist MK-801 (Figure 2G). When these assays were performed in the presence of α-syn fibrils, the signals were similar to those obtained in the absence of fibrils (Figure 2F,H).

### 2.3. α-Syn Fibrils Decrease CB_1_R-NMDAR Signaling in a Heterologous System

Then, we studied the functionality of the CB_1_R-NMDAR heteromer in the presence of α-syn fibrils. When HEK-293T cells expressing CB_1_R and NMDAR were stimulated with forskolin and were treated with the selective CB_1_R agonist, ACEA, a decrease in cAMP levels (around 30%) was observed. However, treatment with NMDA was not able to modify cAMP levels (Figure 3A). When cells were co-treated with ACEA and NMDA, a decrease in cAMP levels similar to those obtained with ACEA treatment alone was observed, indicating that NMDAR activation does not affect CB_1_R signaling (Figure 3A). When the cells were pre-treated with antagonists, the CB_1_R-induced signal was blocked not only by rimonabant but also by the NMDAR antagonist MK-801 (Figure 3A). This phenomenon is known as cross-antagonism and was previously described in this heteromer by Navarro et al. [25]. Surprisingly, when this assay was performed in the presence of α-syn fibrils, ACEA was not able to diminish cAMP levels (Figure 3B).

Next, we studied the Ca^2+^ signaling pathway in HEK-293T cells expressing CB_1_R and NMDAR. When these cells were treated with NMDA, an increase in Ca^2+^ levels was detected, but when cells were co-activated with ACEA and NMDA, the signal was significantly diminished, indicating a negative cross-talk phenomenon between CB_1_R and NMDAR (Figure 3C). The NMDA-induced signal was blocked not only by pre-incubation with the NMDAR antagonist MK-801 but also with the CB_1_R antagonist rimonabant (Figure 3C), showing a cross-antagonism from CB_1_R to NMDAR. When cells were pre-treated with α-syn fibrils, similar results were observed (Figure 3D).

Since another of the signaling pathways activated by NMDAR is MAPK, we analyzed ERK1/2 phosphorylation. In HEK-293T cells expressing CB_1_R and NMDAR, treatment with the CB_1_R agonist, ACEA, or with NMDA produced a significant increase in ERK1/2 phosphorylation levels (Figure 3E). When these cells were co-treated with both agonists, a non-additive effect was observed (Figure 3E), as the signal obtained was lower than the expected signal resulting from the summation of the signals from the individual treatments. The signals of both ACEA and NMDA were blocked by the antagonist of the partner receptor (Figure 3E), and, thus, a bidirectional cross-antagonism characterizes this complex. When this assay was performed in the presence of α-syn fibrils, the non-additive effect and the bidirectional cross-antagonism were still present, but the signals of ACEA and NMDA were smaller than in the control cells (Figure 3F).

Altogether, these results show that α-syn fibrils induce a decrease in the signaling of both CB_1_R and NMDAR in the CB_1_R-NMDAR heteromer in transfected HEK-293T cells.

### 2.4. α-Syn Fibrils Reduce CB_1_R-NMDAR Heteromer Expression in Striatal Neurons

After studying the functionality of the CB_1_R-NMDAR heteromer in a heterologous system, we studied the presence of this receptor complex in striatal neurons, as the striatum is one of the brain areas most affected in Parkinson’s disease. Thus, neurons were treated with 10 μg/L of human α-syn fibrils for 48 h as an in vitro PD model. Immunocytochemistry with an antibody detecting human α-syn showed that neurons treated with α-syn fibrils are capable of incorporating human α-syn (Figure 4A).

To detect the expression of CB_1_R-NMDAR heteromer in neurons, a Proximity Ligation Assay (PLA) was carried out in the primary cultures of rat striatal neurons. The presence of red fluorescent dots indicated that CB_1_R and NMDAR form heteromers in these neurons, with around 70 dots/cell with dots, versus the negative control condition, in which the anti-CB_1_R primary antibody was omitted, and which showed around 5 dots/cell with dots (Figure 4B). The treatment with α-syn fibrils caused a decrease of approximately 50 % in the expression of the heteromer, as these neurons presented around 37 dots/cell with dots (Figure 4B).

### 2.5. CB_1_R Activation Protects Neurons against α-Syn Fibrils-Induced Neuronal Death

When the effect of α-syn fibrils on neuronal viability was studied, it was observed that α-syn fibrils produced a 35 ± 10% decrease in cell viability (Figure 4C). However, the treatment with the CB_1_R agonist, ACEA, counteracted this effect, while the treatment with NMDA was not able to protect cells (Figure 4C). This result suggests that ACEA is able to protect striatal neurons against cell death induced by α-syn fibrils.

### 2.6. The Functionality of CB_1_R-NMDAR Complexes Is Decreased by α-Syn Fibrils in Striatal Neurons

Next, we studied how α-syn fibrils affect the functionality of CB_1_R-NMDAR heteromer in primary cultures of striatal neurons. When the intracellular cAMP pathway was analyzed, it was observed that the CB_1_R agonist, ACEA, was able to decrease cAMP levels previously increased with forskolin, while NMDA was not able to change cAMP levels (Figure 4D). Co-treatment with ACEA and NMDA produced a decrease similar to that of ACEA alone. When neurons were pre-treated with the selective antagonists, it was observed that the effect of ACEA was counteracted not only by the CB_1_R antagonist rimonabant but also by the treatment with NMDAR antagonist MK-801 (Figure 4D), indicating that there is a cross-antagonism effect from NMDAR to CB_1_R. When neurons were pre-treated with α-syn fibrils, the cAMP decrease produced upon ACEA treatment was smaller compared to cells treated with the vehicle (Figure 4E). On the other hand, ERK1/2 phosphorylation was also analyzed in striatal neurons. When cells were treated with ACEA or with NMDA, both agonists were able to produce an increase in pERK1/2 levels, and again the non-additive effect and the bidirectional cross-antagonism were detected (Figure 4F). When neurons were pre-treated with α-syn fibrils, the effects of both agonists were smaller compared to the cells treated with the vehicle (Figure 4G). These results agree with the observations in transfected HEK-293T cells treated with α-syn fibrils.

## 3. Discussion

One of the hallmarks of Parkinson’s disease is the intracellular inclusions known as Lewy bodies, which are composed of misfolded and aggregated α-synuclein [5]. It is well known that oligomeric α-synuclein is toxic in vivo [40]; α-synucleinopathy can be observed in in vitro and in vivo models after the intracerebral inoculation of pathological α-syn seeds in the form of preformed fibrils (PFFs) [41]. Both in vivo and in vitro studies show rapid uptake of α-synuclein fibrils into neurons [42,43,44,45], and it has been shown that α-synuclein is internalized by neurons via endocytosis [46]. Here, we observed a decrease in the signaling of both CB_1_ and NMDA receptors in CB_1_R-NMDAR complexes, as well as a reduction in the expression of these complexes, in an in vitro model of Parkinson’s disease consisting of primary cultures of striatal neurons treated with preformed fibrils of α-synuclein.

The implication of NMDARs in PD pathology is clear, as glutamate, an excitatory neurotransmitter, plays a key role in the disruption of physiological basal ganglia function. α-syn has been described to induce cognitive impairment through GluN2B-containing NMDAR overactivation by means of a mechanism involving the cellular prion protein (PrPC), the metabotropic glutamate receptors 5 (mGluR5), and Fyn kinase [47]. Also, other NMDAR subunits seem to be implicated in α-syn oligomers-induced effects. The loss of LTP and motor and cognitive defects in in vivo PD models was mechanistically attributed to α-syn oligomers directly targeting GluN2D-containing NMDARs [48]. In the striatum, GluN2A localization at the postsynaptic site was disturbed by α-syn oligomers, which caused a reduction in NMDA receptor-mediated synaptic currents and an impairment in corticostriatal and thalamostriatal LTP in spiny projection neurons of both direct and indirect pathways in vitro [49]. Of interest, monoclonal antibodies against oligomeric forms of α-syn were able to prevent the reduction of GluN2A levels in the postsynaptic compartment and the consequent loss of LTP [49]. Furthermore, in vivo intrastriatal injection of α-syn oligomers in experimental animals led to deficits in visuospatial learning, in association with the reduced expression of GluN2A NMDA receptor subunit, indicating a selective targeting by α-syn oligomers both ex vivo and in vivo [49]. Therefore, this evidence, together with the alterations in NMDAR subunits’ expression described in PD animal models [14,18,19], suggests that NMDA receptors are interesting targets for the treatment of PD.

When analyzing NMDA-induced Ca^2+^ signals in a heterologous expression system expressing either NMDAR alone or CB_1_R and NMDAR, we observed that, upon α-synuclein treatment, the signal was similar to that obtained in cells treated with the vehicle. This may seem to disagree with the literature, as it has been described that exposure of neurons to α-syn oligomers increases intracellular Ca^2+^ levels [47,50,51]. However, this difference might be explained by the fact that HEK-293T cells do not express the proteins that are involved in that α-syn-mediated intracellular Ca^2+^ increase, such as PrPC, Fyn kinase, and mGluR5 [47], or Cav2.2 Channels [50]; thus, the involved mechanisms cannot take place in HEK-293T cells.

When we analyzed the NMDA-induced signal in the MAPK phosphorylation pathway, we observed that, upon α-syn fibrils treatment, this signal was decreased in primary cultures of neurons. This might be due to a response mechanism from the cell that tunes down NMDAR activation to reduce the toxicity produced by α-synuclein.

We also analyzed CB_1_R-mediated signaling, and both in transfected HEK-293T cells and in primary cultures of neurons, we observed a decrease in the signal induced by ACEA upon α-syn fibrils treatment. These results might be explained by the alterations in the CB_1_R expression that have been described in the early stages of PD. A reduction in CB_1_R expression was found in early-stage PD patients [52] and in early-phase genetic PD animal models [53]. However, the expression of CB_1_R varies along the course of PD progression, as an up-regulation of CB_1_Rs has been found in intermediate and advanced stages of PD. Furthermore, overactivity of CB_1_R during late stages of PD has been shown in rodents and primates lesioned with neurotoxins [54,55] and in mice with mutations [53].

Regarding the necessity for better treatments for PD, interestingly, NMDA receptor antagonists have shown effective antiparkinsonian effects in animal models of PD, and they can reduce the complications of chronic dopaminergic therapy, such as wearing off and dyskinesias [56,57,58]. These results highlight the detrimental role of NMDAR activation in PD and point to NMDARs as new targets for improving the therapeutic strategies used to treat Parkinson’s disease. However, the use of these agents in humans has been limited due to the adverse effects associated with nonselective blockade of NMDA receptor function [59], but the development of more potent and selective compounds holds the promise of an important new therapeutic approach for PD. Here, we observed that when both receptors in the CB_1_R-NMDAR heteromer were activated, the Ca^2+^ signal was lower than the signal induced by NMDA alone, as previously described by Navarro and collaborators [25]. The fact that CB_1_R’s activation is able to decrease NMDAR-induced signal points to a neuroprotective role of CB_1_R in the CB_1_R-NMDAR complex. Moreover, given the neuroprotective effect of cannabinoids [34], CB_1_R would have a double-beneficial effect by both providing cannabinoid signaling and by reducing NMDAR activation. Thus, the CB_1_R-NMDAR complex appears as a new therapeutic target for the treatment of PD. As our results show that the expression of CB_1_R-NMDAR complexes is decreased in in vitro and in vivo PD models, we speculate that favoring the expression or the functionality of this receptor heteromer could be beneficial in a PD context. This way, cannabinoids would be interesting new therapeutic agents for the treatment of PD, as they could not only provide neuroprotective effects but also decrease NMDAR-induced excitotoxicity, thanks to the reduction of NMDAR signaling induced by CB_1_R activation in CB_1_R-NMDAR complexes. Cannabinoids represent a different therapeutic strategy from the use of NMDAR antagonists that would avoid the adverse effects associated with nonselective blockade of NMDA receptor function. More research and clinical trials with cannabinoids lacking psychoactive effects are necessary to find better treatments for Parkinson’s disease.

## 4. Materials and Methods

### 4.1. Drugs

ACEA (#1319), N-Methyl-d-aspartate (NMDA) (#0114), SR 141716A (rimonabant hydrochloride) (#0923), (+)-MK 801 maleate (MK) (#0924), and zardaverine (#1046) were purchased from Tocris (Bristol, UK).

Forskolin (FK) (#HY-15371/CS-1454) was purchased from MedChemExpress (Monmouth Junction, NJ, USA).

### 4.2. Cell Culture and Transient Transfection

HEK-293T cells at passage 8–12 were grown in Dulbecco’s modified Eagle’s medium (DMEM) (15-013-CV, Corning Inc., Corning, NY, USA) supplemented with 2 mM L-glutamine, 100 U/mL penicillin/streptomycin, MEM Non-Essential Amino Acids Solution (1/100), and 5% (*v*/*v*) heat-inactivated fetal bovine serum (FBS) (Invitrogen, Paisley, Scotland, UK). Cells were maintained in a humid atmosphere of 5% CO_2_ at 37 °C. Briefly, HEK-293T cells growing in 6-well dishes or in 25 cm2 flasks were transiently transfected using the PEI (PolyEthylenImine, Sigma-Aldrich, St. Louis, MO, USA) method. Cells were incubated for 4 h with the corresponding cDNAs, together with PEI (5.47 mM in nitrogen residues) and 150 mM NaCl in a serum-starved medium. Then, the medium was replaced by a fresh complete culture medium, and cells were incubated for 48 h before experimental procedures.

To prepare primary neuronal cultures, the brain from Sprague Dawley rat embryos (E19) was removed. The striatum was dissected and carefully stripped off the meninges. Tissue was processed as described above for microglial cultures, except that neurons were grown in a neurobasal medium (21103-049, Gibco, Paisley, Scotland, UK) supplemented with 2 mM L-glutamine, 100 U/mL penicillin/streptomycin, MEM non-essential amino acids preparation (1/100), and 2% (*v*/*v*) B27 supplement (17504-044, Gibco, Paisley, Scotland, UK). Cultures were maintained at 37 °C in a humidified 5% CO_2_ atmosphere for 12 days.

### 4.3. Fusion Proteins and Expression Vectors

The human cDNAs for the CB_1_ and GHS-R_1_a receptors and NR1A and NR2B NMDAR subunits cloned in pcDNA3.1 were amplified without their stop codons, using sense and antisense primers. The primers harbored either unique BamHI and KpnI sites for CB_1_R, EcoRI and KpnI sites for GHS-R_1_a, or BamHI and HindIII sites for NR1A. The fragments were subcloned to be in frame with an enhanced yellow fluorescent protein (pEYFP-N1; Clontech, Heidelberg, Germany) or an Rluc (pRluc-N1; PerkinElmer, Wellesley, MA, USA) on the C-terminal end of the receptor to produce NR1A-Rluc, CB_1_R–YFP and GHS-R_1_a-YFP fusion proteins.

### 4.4. α-Synuclein Treatment

HEK-293T or primary neuronal cell cultures were treated for 48 h with recombinant human α-synuclein fibrils obtained via sonication, at a final concentration of 10 µg/L. Fibrils were prepared as previously described [60,61].

### 4.5. Cell Viability

Cell viability assay is based on the principle that living cells maintain intact cell membranes that exclude certain dyes, like trypan blue. To quantify the percentage of living cells, DIV 14 primary cultures of striatal neurons growing in 6-well plates were treated with 10 µg/L α-syn or vehicle. On DIV 15, cells were treated either with 100 nM ACEA, 15 μM NMDA, or vehicle for another 24 h.

On the day of the experiment, cells were gently detached and mixed with an equal volume of trypan blue (0.4%) (Trypan Blue solution, T8154, Sigma Aldrich (St. Louis, MO, USA)). Cells (%) were counted in a Countess II FL automated cell counter (Thermo Fisher Scientific, Waltham, MA, USA).

### 4.6. Bioluminescence Resonance Energy Transfer (BRET) Assays

HEK-293T cells growing in 6-well plates were transiently co-transfected with a constant amount of cDNA encoding for NR1A fused to Renilla luciferase (NR1A-Rluc), with a constant amount of the cDNA encoding for NR2B, and with increasing amounts of cDNAs corresponding to CB_1_R or ghrelin receptor GHSR_1a_ fused to the yellow fluorescent protein (CB_1_-YFP and GHSR_1a_-YFP). Forty-eight hours post-transfection, cells were washed twice in quick succession in HBSS (137 mM NaCl; 5 mM KCl; 0.34 mM Na_2_HPO_4_; 0.44 mM KH_2_PO_4_; 1.26 mM CaCl_2_; 0.4 mM MgSO_4_; 0.5 mM MgCl_2_ and 10 mM HEPES, pH 7.4) supplemented with 0.1% glucose (*w*/*v*), detached by gently pipetting and resuspended in the same buffer. To assess the number of cells per plate, we determined the protein concentration using a Bradford assay kit (Bio-Rad, Munich, Germany) with bovine serum albumin dilutions as standards. To quantify YFP-fluorescence expression, we distributed the cells (20 μg protein) in 96-well microplates (black plates with a transparent bottom; Porvair, Leatherhead, UK). Fluorescence was read using a Mithras LB 940 (Berthold, Bad Wildbad, Germany) equipped with a high-energy xenon flash lamp, using a 10 nm bandwidth excitation and emission filters at 485 and 530 nm, respectively. YFP-fluorescence expression was determined as the fluorescence of the sample minus the fluorescence of cells expressing protein–Rluc alone. For the BRET measurements, the equivalent of 20 μg of cell suspension was distributed in 96-well microplates (white plates; Porvair), and we added 5 μM coelenterazine H (PJK GMBH, Kleinblittersdorf, Germany). Then, 1 min after coelenterazine H addition, the readings were collected using a Mithras LB 940 (Berthold, Bad Wildbad, Germany), which allowed for the integration of the signals detected in the short-wavelength filter at 485 nm (440–500 nm) and the long-wavelength filter at 530 nm (510–590 nm). To quantify receptor–Rluc expression, we performed luminescence readings 10 min after the addition of 5 μM coelenterazine H. The net BRET is defined as [(long-wavelength emission)/(short-wavelength emission)]-Cf, where Cf corresponds to [(long-wavelength emission)/(short-wavelength emission)] for the Rluc construct expressed alone in the same experiment. The BRET curves were fitted assuming a single phase by a non-linear regression equation, using the version 9.5.0 (525) of GraphPad Prism software (San Diego, CA, USA). BRET values are given as milli BRET units (mBU: 1000× net BRET).

### 4.7. cAMP Level Determination

Two hours before initiating the experiment, HEK-293T or neuronal cell-culture medium was exchanged to serum-starved DMEM or neurobasal medium, as corresponds. Then, cells were detached, resuspended in the serum-starved medium containing 50 µM zardaverine, plated in 384-well microplates (2500 cells/well), pre-treated (15 min) with the corresponding antagonists or the vehicle, and then stimulated with agonists (15 min) before adding 0.5 μM forskolin or vehicle. Readings were performed after 1 h incubation at 25 °C. Homogeneous time-resolved fluorescence energy transfer (HTRF) measures were obtained using the Lance Ultra cAMP kit (PerkinElmer, Waltham, MA, USA) [62]. Fluorescence at 665 nm was analyzed on a PHERAstar Flagship microplate reader equipped with an HTRF optical module (BMG Lab technologies, Offenburg, Germany).

### 4.8. Extracellular Signal-Regulated Kinase Phosphorylation Assays

HEK-293T cells growing in 25 cm^2^ flasks were transfected with the cDNAs encoding for CB_1_R, NR1A, and NR2B. Two-to-four hours before initiating the experiment, the culture medium was replaced by serum-starved DMEM medium. Cells were incubated at 37 °C with antagonists (15 min) or the vehicle, followed by stimulation (7 min) with agonists. After that, the reaction was stopped by placing cells on ice. Then, cells were washed twice with cold PBS and lysed by the addition of ice-cold lysis buffer (50 mM Tris-HCl pH 7.4, 50 mM NaF, 150 mM NaCl, 45 mM glycerol-3-phosphate, 1% Triton X-100, 20 µM phenyl-arsine oxide, 0.4 mM NaVO_4_, and protease inhibitor mixture (MERK, St. Louis, MO, USA)) Cellular debris were removed by centrifugation at 13,000× *g* for 10 min at 4 °C, and protein concentration was adjusted to 1 mg/mL by the bicinchoninic acid method (ThermoFisher Scientific, Waltham, MA, USA), using a commercial bovine serum albumin dilution (BSA) (ThermoFisher Scientific, Waltham, MA, USA) for standardization. Then, 6× Laemmli SDS sample buffer (300 mM Tris-Base, 600 mM DTT, 40% glycerol (*v*/*v*), 0.012% Bromophenol blue (*w*/*v*), and 12% SDS (*w*/*v*), pH = 6.8) were added to the samples, and proteins were denatured by boiling at 100 °C for 5 min. ERK1/2 phosphorylation was determined by Western blot. Equivalent amounts of protein (20 μg) were subjected to electrophoresis (10% SDS–polyacrylamide gel) and transferred onto PVDF membranes (Immobilon-FL PVDF membrane, MERK, St. Louis, MO, USA) for 30 min, using Trans-Blot Turbo system (Bio-Rad). Then, the membranes were blocked for 2 h at room temperature (constant shaking) with Odyssey Blocking Buffer (LI-COR Biosciences, Lincoln, NE, USA) and labeled with a mix of primary mouse anti-phospho-ERK 1/2 (1/2500, MERK, Ref. M8159) and rabbit anti-ERK 1/2 (1/40,000, MERK, Ref. M5670) antibodies overnight at 4 °C, with shaking. Then, the membranes were washed three times with PBS containing 0.05% tween for 10 min and subsequently were incubated with a mix of IRDye 800 anti-mouse (1/10,000, MERK, Ref. 92632210) and IRDye 680 anti-rabbit (1/10,000, MERK, Ref. 926-68071) secondary antibodies for 2 h at room temperature, light protected. Membranes were washed 3 times with PBS–tween 0.05% for 10 min and once with PBS and left to dry. Bands were analyzed using Odyssey infrared scanner (LI-COR Biosciences, Lincoln, NE, USA). Band densities were quantified using version 2.14.0/1.54f of Fiji software, and the level of phosphorylated ERK1/2 was normalized using the total ERK 1/2 protein band intensities. Results are represented as the percentage over basal (non-stimulated cells).

To determine extracellular signal-regulated kinase 1/2 (ERK1/2) phosphorylation in neuronal and glial primary cultures, cells were grown in 96-well plates. On the day of the experiment, the medium was replaced by serum-free medium 2 h before starting the experiment. The cells were pre-treated at 25 °C for 15 min with antagonists or the vehicle and stimulated for an additional 15 min with selective agonists. Cells were then washed twice with cold PBS before the addition of lysis buffer (a 15 min treatment). Afterward, 10 µL of each supernatant was placed in white ProxiPlate 384-well plates, and ERK 1/2 phosphorylation was determined using an AlphaScreen^®^SureFire^®^ kit (Perkin Elmer, Waltham, MA, USA), following the instructions of the supplier, and readings were collected using an EnSpire^®^ Multimode Plate Reader (PerkinElmer, Waltham, MA, USA). The value of reference (100%) was the value achieved in the absence of any treatment (basal). The effect of ligands was given in percentage with respect to the basal value.

### 4.9. Detection of Cytoplasmic Calcium Levels

HEK-293T cells were cotransfected with the cDNA for the corresponding receptors (see figure legend) together with the cDNA for the GCaMP6 calcium sensor [63]. Forty-eight hours after transfection, HEK-293T cells were detached using Mg^2+^-free Locke’s buffer (154 mM NaCl, 5.6 mM KCl, 3.6 mM NaHCO_3_, 2.3 mM CaCl_2_, 5.6 mM glucose, 5 mM HEPES, 10 μM glycine, pH 7.4), centrifuged for 5 min at 3200 rpm, and resuspended in the same buffer. Protein concentration was quantified by using the Bradford assay kit (Bio-Rad, Munich, Germany). To measure Ca^2+^ mobilization, cells (40 µg of protein) were distributed in 96-well microplates (black plates with a transparent bottom; Porvair, Leatherhead, UK) and were incubated for 10 min with antagonists when indicated. Fluorescence readings were performed right after the addition of agonists. Fluorescence emission intensity due to GCaMP6 was recorded at 515 nm upon excitation at 488 nm on the EnSpire^®^ Multimode Plate Reader for 300 s every 5 s at 100 flashes per well.

### 4.10. Immunofluorescence Studies

HEK-293T neurons growing on glass coverslips were fixed in 4% paraformaldehyde for 15 min and then washed twice with PBS containing 20 mM glycine before permeabilization with the same buffer containing 0.2% Triton X-100 (5 min incubation). Cells were treated for 1 h with PBS containing 1% bovine serum albumin. To detect the expression of NR1-Rluc, cells were labeled with a mouse anti-Rluc antibody (1/100; MAB4400, Millipore, Burlington, MA, USA) and subsequently treated with Cy3-conjugated anti-mouse IgG secondary antibody (1/200; 715-166-150; Jackson ImmunoResearch, West Grove, PA, USA) (1 h each). The expression of CB_1_R-YFP was detected by the YFP’s own fluorescence. The presence of α-syn fibrils was detected with a mouse monoclonal anti-human α-synuclein antibody (1/300; ab1903, Abcam). Phalloidin was detected with an AF488 pre-stained anti-phalloidin probe (1/200; A12379, ThermoFisher, Waltham, MA, USA). Nuclei were stained with Hoechst33432 (1/100 from stock 1 mg/mL; Thermo Fisher, Waltham, MA, USA). The samples were washed several times and mounted on glass slides with ShandonTM Immu-MountTM (9990402; ThermoFisher, Waltham, MA, USA). Samples were observed under a Zeiss 880 confocal microscope (Carl Zeiss, Oberkochen, Germany) equipped with an apochromatic 63× oil-immersion objective (N.A. 1.4) and with 405 nm, 488 nm, and 561 nm laser lines.

### 4.11. In Situ and In Vitro Proximity Ligation Assay (PLA)

The Proximity Ligation Assay (PLA) allows the detection of molecular interactions between two endogenous proteins ex vivo. PLA requires both receptors to be sufficiently close (<16 nm) to allow for the double-strand formation of the complementary DNA probes conjugated to the antibodies. Using the PLA, the heteromerization of NR1 subunits of NMDAR with CB_1_ receptors was detected in situ in primary cultures of neurons and in rat brain sections.

The presence/absence of receptor–receptor molecular interactions in the samples was detected using the Duolink II In Situ PLA Detection Kit (developed by Olink Bioscience, Uppsala, Sweden; and now distributed by Sigma-Aldrich as Duolink^®^, using PLA^®^ Technology). The PLA probes were obtained after conjugation of the primary anti-NR1 antibody (ab52177, Abcam, Cambridge, UK) to a MINUS oligonucleotide (DUO92010, Sigma-Aldrich, St. Louis, MO, USA), and the anti-CB_1_R (ab259323, Abcam, Cambridge, UK) antibody to a PLUS oligonucleotide (DUO92009, Sigma-Aldrich, St. Louis, MO, USA). The specificity of antibodies was tested in non-transfected HEK-293T cells. Samples were fixed in 4% paraformaldehyde for 15 min and then washed twice with PBS containing 20 mM glycine before permeabilization with PBS–glycine containing 0.2% Triton X-100 for 5 min. After permeabilization, the samples were washed in PBS at room temperature and incubated in a preheated humidity chamber for 1 h at 37 °C, with the blocking solution provided in the PLA kit. Then, the samples were incubated overnight with the PLA probe-linked antibodies (1/100 dilution for all antibodies) at 4 °C. After washing, the samples were incubated with the ligation solution for 1 h and then washed and subsequently incubated with the amplification solution for 100 min (both steps at 37 °C in a humid chamber). The expression of MAP2 was detected with a mouse anti-MAP2 primary antibody (1/300, M4403, Sigma-Aldrich, St. Louis, MO, USA), followed by an AF488-conjugated anti-mouse secondary antibody (1/300, A11017, Thermo Fisher Scientific-Life Technologies, Waltham, MA, USA). Nuclei were stained with Hoechst33432 (1/100 from stock 1 mg/mL; Thermo Fisher, Waltham, MA, USA). The samples were washed several times and mounted on glass slides with ShandonTM Immu-MountTM (9990402; ThermoFisher, Waltham, MA, USA).

Negative controls were performed by omitting the anti-CB_1_R-PLUS antibody. Samples were observed under a Zeiss 880 confocal microscope (Carl Zeiss, Oberkochen, Germany) equipped with an apochromatic 63× oil-immersion objective (N.A. 1.4) and with 405 nm, 488 nm, and 561 nm laser lines. For each field of view, a stack of three channels (one per staining) and 9 Z planes with a step size of 0.5 µm were acquired. The ratio, r (number of red spots/cell), was determined on the maximum projection of each image stack, using the version 1.0.1.2 of the Duolink Image tool software.

### 4.12. Data Analysis

Data, expressed as the mean ± SEM, were obtained from at least five independent experiments. Data comparisons were analyzed by one-way ANOVA or two-way ANOVA, followed by Bonferroni’s post hoc test. The normality of populations and homogeneity of variances were tested before the ANOVA. Statistical analysis was undertaken only when each group size was at least n = 5, with n being the number of independent variables (technical replicates were not treated as independent variables). Differences were considered significant when *p* ≤ 0.05. Statistical analyses were carried out with GraphPad Prism software version 9 (San Diego, CA, USA). Outliers’ tests were not used, and all data points (mean of replicates) were used for the analyses.

## Figures and Tables

**Figure 1 ijms-25-03021-f001:**
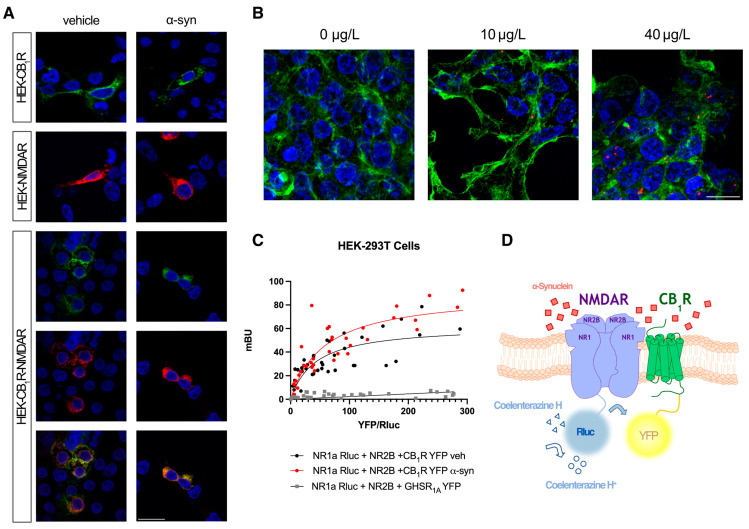
Analysis of CB_1_R-NMDAR complex formation in the presence of α-syn. (**A**) Immunocytochemistry assay was performed in HEK-293T cells treated or not with 10 μg/L of human α-syn fibrils for 48 h and transfected with either CB_1_R-YFP (1 µg cDNA) (shown in green), NR1-Rluc (0.75 µg cDNA) (shown in red) plus NR2B (0.75 µg cDNA), or both. Nuclei were stained with Hoechst (blue). Colocalization is shown in yellow. Scale bar: 15 µm. (**B**) Immunocytochemistry assay was performed in HEK-293T cells treated or not with 10 or 40 μg/L of human α-syn fibrils for 48 h. α-syn fibrils were detected with a mouse anti-human α-synuclein antibody (red). Phalloidin was detected with an AF488 pre-stained anti-phalloidin probe (green). Nuclei were stained with Hoechst (blue). Scale bar: 15 µm. (**C**) BRET assays were performed in HEK-293T cells treated or not with 10 μg/L of human α-syn fibrils for 48 h and transfected with constant amounts of cDNAs for NR1-Rluc (0.5 µg) and NR2B (0.3 µg) and increasing amounts of cDNA for CB_1_R-YFP (0 to 2.5 µg) or GHSR_1a_-YFP (0 to 10 µg). Values are the mean ± SEM of 5 different experiments performed in triplicates. (**D**) Scheme of the BRET technique.

**Figure 2 ijms-25-03021-f002:**
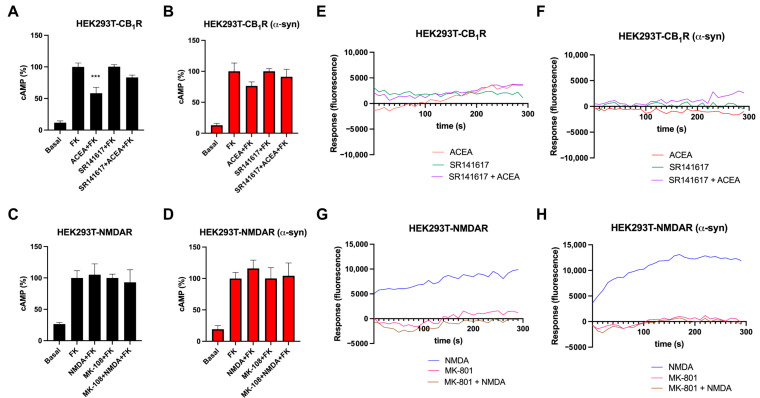
Analysis of CB_1_R and NMDAR signaling in the presence of α-syn fibrils in a heterologous system. (**A**–**D**) HEK-293T cells transfected with CB_1_R (1.5 µg cDNA) (**A**,**B**) or with NR1 (1 µg cDNA) plus NR2B (1 µg cDNA) (**C**,**D**) were treated (**B**,**D**) or not (**A**,**C**) with α-syn fibrils. Forty-eight hours after, cells were pre-treated with the vehicle or with the selective antagonists (1 µM SR141617 for CB_1_R or 1 µM MK-801 for NMDAR), followed by agonist stimulation (100 nM ACEA for CB_1_R or 15 µM NMDA for NMDAR). cAMP accumulation was detected by HTRF in the presence of 0.5 µM forskolin. Values are the mean ± SEM of 6 different experiments performed in triplicates, and one-way ANOVA, followed by Tukey’s multiple comparison post hoc test, was used for statistical analysis (*** *p* < 0.001; versus treatment with forskolin). (**E**–**H**) Calcium release was evaluated in HEK-293T transfected with CB_1_R (1 µg cDNA) (**E**,**F**) or with NR1 (1 µg cDNA) plus NR2B (1 µg cDNA) (**G**,**H**) and with 6GCamMP calcium sensor (0.75 µg cDNA) and treated (**F**,**H**) or not (**E**,**G**) with α-syn fibrils. Cells were pre-treated with the vehicle or with the selective antagonists (1 µM SR141617 for CB_1_R or 1 µM MK-801 for NMDAR), followed by agonist stimulation (100 nM ACEA for CB_1_R or 15 µM NMDA for NMDAR). Data represent the mean ± SEM of six-to-eight different experiments performed in triplicates.

**Figure 3 ijms-25-03021-f003:**
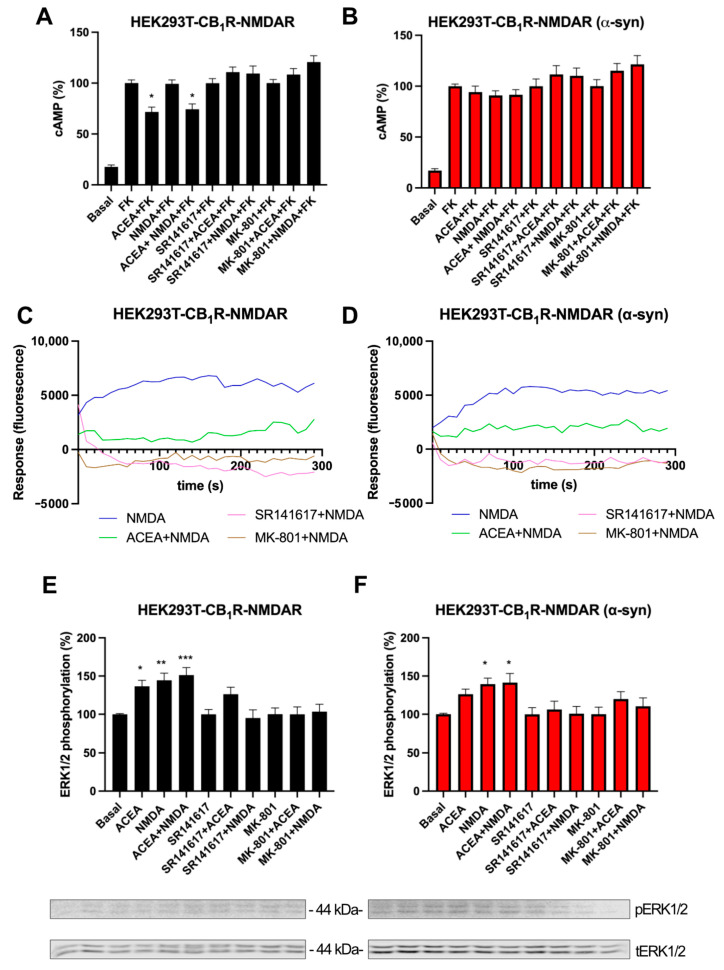
Functional analysis of CB_1_R-NMDAR complexes in a heterologous system upon α-syn treatment. (**A**,**B**) HEK-293T cells were transfected with the cDNAs for CB_1_R (0.5 µg), NR1 (1 µg), and NR2B (1 µg) and treated (**B**) or not (**A**) with α-syn fibrils. Cells were activated with the selective antagonists (1 µM SR141617 for CB_1_R or 1 µM MK-801 for NMDAR), followed by agonist stimulation (100 nM ACEA for CB_1_R and/or 15 µM NMDA for NMDAR). cAMP accumulation was detected by HTRF in the presence of 0.5 µM forskolin. (**C**,**D**) Calcium release was evaluated in HEK-293T cells transfected with the cDNAs for CB_1_R (0.75 µg), NR1 (0.75 µg) NR2B (0.75 µg), and 6GCamMP calcium sensor (0.75 µg) and treated (**D**) or not (**C**) with α-syn fibrils. Cells were activated with the selective antagonists, followed by agonist stimulation. (**E**,**F**) HEK-293T cells were transfected with the cDNAs for CB_1_R (0.5 µg), NR1 (1 µg), and NR2B (1 µg) and treated (**F**) or not (**E**) with α-syn fibrils. Cells were activated with the selective antagonists (1 µM SR141617 for CB_1_R or 1 µM MK-801 for NMDAR), followed by agonist stimulation (100 nM ACEA for CB_1_R and/or 15 µM NMDA for NMDAR), and MAPK phosphorylation was detected by Western blot (blots for phosphorylated and total ERK1/2 (42–44 kDa) are shown below). Data represent the mean ± SEM of six-to-eight different experiments performed in triplicates. One-way ANOVA and Tukey’s multiple comparison post hoc test were used for statistical analysis (* *p* < 0.05, ** *p* < 0.01, and *** *p* < 0.001; versus forskolin in cAMP or versus basal in MAPK).

**Figure 4 ijms-25-03021-f004:**
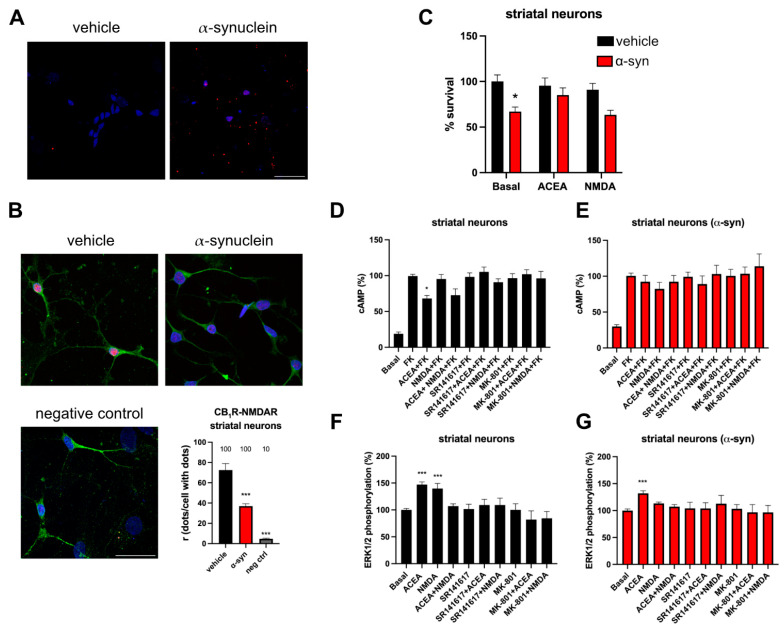
Expression and functionality of CB_1_R-NMDAR complexes in striatal neurons treated with α-syn fibrils. (**A**) Primary cultures of rat striatal neurons were treated with α-syn or vehicle. α-syn fibrils were detected by immunocytochemistry with a mouse anti-human α-synuclein antibody (red). Nuclei were stained with Hoechst (blue). Scale bar: 30 µm. (**B**) Proximity Ligation Assay (PLA) was performed in primary cultures of rat striatal neurons treated or not with α-syn fibrils. Confocal microscopy images are shown (superimposed sections) in which heteromers appear as red clusters. Neurons were labeled using a mouse anti-MAP2 antibody (green). Cell nuclei were stained with Hoechst (blue). Scale bar: 30 μm. Quantification of the number of red dots/cells with dots (r) and of the percentage of cells presenting red dots is shown. Values are the mean ± SEM (n = 6). One-way ANOVA, followed by Tukey’s multiple comparison post hoc test, was used for statistical analysis (*** *p* < 0.001, versus vehicle). (**C**) Primary cultures of rat striatal neurons were treated with α-syn or vehicle for 24 h, followed by treatment with ACEA (100 nM) or NMDA (15 mM) or vehicle for another 24 h period. Then, cells were gently detached, and neuronal survival was assessed with a cell counter. Values are the mean ± SEM of 5 independent experiments performed in triplicates. Two-way ANOVA, followed by Tukey’s multiple comparison post hoc test, was used for statistical analysis (* *p* < 0.05, versus vehicle condition). (**D**–**G**) Primary cultures of rat striatal neurons treated (**E**,**G**) or not (**D**,**F**) with α-syn were stimulated with the selective antagonists (1 µM SR141617 for CB_1_R or 1 µM MK-801 for NMDAR), followed by agonist stimulation (100 nM ACEA for CB_1_R and/or 15 µM NMDA for NMDAR) and cAMP levels (**D**,**E**), and MAPK phosphorylation signals were measured (**F**,**G**). Values are the mean ± SEM of 6 different experiments. One-way ANOVA, followed by Tukey’s multiple comparison post hoc test, were used for statistical analysis (* *p* < 0.05, *** *p* < 0.001; versus forskolin in cAMP assay or versus basal in MAPK phosphorylation).

## Data Availability

Data can be obtained from the corresponding author upon reasonable request.

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
