# Peer review of "The Expression and Functionality of CB1R-NMDAR Complexes Are Decreased in A Parkinson’s Disease Model"

_ijms, 2024, doi:10.3390/ijms25053021_

Round 1

Reviewer 1 Report

Comments and Suggestions for Authors

The manuscript by Reyes-Resian et al. focuses on studying the contribution of alpha-synuclein treatment to the interaction between CB1R and NMDAR in both HEK cells, neurons, and a PD disease model. While the authors provide some interesting observations, without further explanation, it is difficult for the audience to fully understand the manuscript.

Major:

The result of α-syn fibrils causing a reduction in CB1R activation in HEK cells and striatal neurons is quite interesting. Firstly, does the reduction of CB1 activation only occur in G proteins or beta-arrestin signaling as well? If the reduction only occurs in G protein signaling (cAMP), what causes this—reduction of G proteins or some other mechanisms? These points should be addressed.

Pre-incubation of the CB1R antagonist rimonabant can block NMDA-induced signaling. This should be further confirmed with other CB1R antagonists instead of claiming cross-antagonism. Additionally, how does CB1R alone (G protein vs beta-arrestin bias agonists) modulate NMDAR function?

The CB1R agonist ACEA can protect striatal neurons from α-syn fibril-induced neuronal death. Is this mediation by G proteins or arrestin signaling, or is it mediated by heterodimerization between CB1R and NMDAR?

The expression of the CB1R-NMDAR heteromer is reduced in PD rats. Firstly, how long were the rats treated with 6-OHDA, and will the earlier phase (a few days post 6-OHDA) versus the later phase (a few weeks post 6-OHDA) show differences in heteromer expression? Is there any statistical difference between those lesioned rats regarding the expression of the heteromer? Also, for those L-DOPA treated rats, how does dyskinesia correlate with the expression of the heteromer?

Minor:

There are numerous typos within the manuscript that should be addressed.

Comments on the Quality of English Language

NA

Reviewer 2 Report

Comments and Suggestions for Authors

The current paper discusses the investigation of CB1R-NMDAR complexes in Parkinson's disease (PD) and their potential as therapeutic targets. However, there are several areas that could be improved or clarified

1.    The abstract contains some long, complex sentences that could be broken into simpler ones for clarity. For example, the use of "where" in line 47 might confuse readers about the location or conditions being described.

2.    The abstract heavily uses technical jargon and abbreviations without providing definitions for a general audience. Terms like "BRET assay," "cAMP," "MAPK signaling," and "PLA" might not be understood by readers not familiar with the field

3.    The findings are presented in a somewhat vague manner. It would be beneficial to include more specific data or results to support the claims made, such as quantitative findings from the BRET assay and PLA.

4.    The introduction briefly mentions the role of CB1R-NMDAR complexes in an Alzheimer's disease (AD) model but doesn't explain why this is relevant to PD. A clearer link between these findings and the rationale for the PD study would be helpful

5.    The significance of the findings is not placed in a broader context, making it difficult for non-specialists to understand the impact of the research.

6.    The mechanism by which α-synuclein affects the CB1R-NMDAR complex and signaling pathways is not clearly articulated

7.    While the current study mentions the potential of targeting CB1R-NMDAR complexes in PD therapy, it does not discuss the broader implications or how this approach compares to existing treatments.

8.    The line numbers-45, 46, 47) are included, which is not standard and can be distracting.

9.    The conclusion could be strengthened by more definitively stating the findings rather than just pointing to a potential role

10. The description of the PD model in rats is vague and could be clarified to explain the relevance of the 6-OH-DA lesion model to human PD.

Comments on the Quality of English Language

Minor corrections

Reviewer 3 Report

Comments and Suggestions for Authors

In this work, the authors attempted to clarify the role of the CB1R- NMDAR receptor complex in Parkinson's disease. The authors' studies show that alpha-synuclein can lead to a reorganization of the CB1R-NMDAR complex in transfected HEK- 293T cells. In addition, treatment with alpha-synuclein resulted in decreased cAMP and MAPK signaling of both CB1R and NMDAR not only in transfected cells but also in primary neuronal cultures. These results indicate a role for CB1R-NMDAR complexes as a novel therapeutic target in Parkinson's disease.

The work is very interesting and well argued. I recommend its publication on International Journal of Molecular Sciences after some minor revisions.

Some questions for the authors:

-In this work, since alpha-synuclein was used, I wonder why the interaction with dopamine was not evaluated? Likewise, why was the effect on alpha-synuclein of iron and copper not considered?

-Has the effect of cannabidiol administration on the CB1R-NMDAR receptor complex been evaluated?

Analysing the manuscript in more detail.

1)    Introduction

The introduction is well written, clear and supported by the literature. I would only ask you to comment on the following sentence by giving other examples from literature “Furthermore, in neurons of an animal model of Alzheimer’s disease, the expression of this complex was increased, suggesting that this receptor heteromer could have a role in neurodegenerative diseases”

2)    Results

2.5. CB1R activation protects neurons against a-syn fibrils-induced neuronal death. In this paragraph, the authors should better clarify how CBR1 prevents neuronal death.

     3)   Discussion

In this section, the experiments are clearly commented on and supported by the literature. However, it lacks a final section in which the authors draw conclusions from the work and a section on future perspectives.

Round 2

Reviewer 1 Report

Comments and Suggestions for Authors

The authors should keep the PD disease animal model instead of deleting it. Without it, the story is not good enough for the journal.

Comments on the Quality of English Language

NA

Author Response

Thank you for the comment.

We decided to remove this data because be believe that the manuscript is better without it, as we now prefer to focus only on the effects of alpha-synuclein. Also some of the reviewers had concerns about the model.